# A Role for ER-Beta in the Effects of Low-Density Lipoprotein Cholesterol and 27-Hydroxycholesterol on Breast Cancer Progression: Involvement of the IGF Signalling Pathway?

**DOI:** 10.3390/cells11010094

**Published:** 2021-12-29

**Authors:** Reham M. Mashat, Hanna A. Zielinska, Jeff M. P. Holly, Claire M. Perks

**Affiliations:** IGFs & Metabolic Endocrinology Group, Translational Health Sciences, Bristol Medical School, Learning & Research Building, Southmead Hospital, Bristol BS10 5NB, UK; mashatreham@gmail.com (R.M.M.); hanna.zielinska@hotmail.com (H.A.Z.); jeff.holly@bristol.ac.uk (J.M.P.H.)

**Keywords:** 27-OHC, breast cancer, oestrogen receptor-beta, IGF-I

## Abstract

Cholesterol—in particular, high levels of low-density lipoprotein (LDL) and its metabolite, 27-hydroxycholesterol (27-OHC)—is correlated with increases in the risks of breast cancer and obesity. Although the high expression of LDL/27-OHC has been reported in breast cancer, its effects and mechanism of action remain to be fully elucidated. In this study, we found that the effects of LDL on cell proliferation were mediated by the activation of the cytochrome P450 enzyme, sterol 27 hydroxylase, and cholesterol 27-hydroxylase (CYP27A1) in both ER-α-positive and ER-α-negative breast cancer cells. We found that treatment with 27-OHC only increased cell growth in oestrogen receptor-α (ER-α)-positive breast cancer cells in an ER-α-dependent manner, but, interestingly, the effects of 27-OHC on cell migration and invasion were independent of ER-α. Using ER-α-negative MDA-MB-231 cells, we found that 27-OHC similarly promoted cell invasion and migration, and this was mediated by oestrogen receptor β (ER-β). These results suggest that 27-OHC promotes breast cancer cell proliferation in ER-α-positive breast cancer cells via ER-α, but migration and invasion are mediated via ER-β in ER-α positive and negative cell lines. The addition of LDL/27OHC increased the production of IGF-I and the abundance of IGF-IR in TNBC. We further found that modulating ER-β using an agonist or antagonist increased or decreased, respectively, levels of the IGF-I and EGF receptors in TNBC. The inhibition of the insulin-like growth factor receptor blocked the effects of cholesterol on cell growth and the migration of TNBC. Using TCGA and METABRIC microarray expression data from invasive breast cancer carcinomas, we also observed that higher levels of ER-beta were associated with higher levels of IGF-IR. Thus, this study shows novel evidence that ER-β is central to the effects of LDL/27OHC on invasion, migration, and the IGF and EGF axes. Our data suggest that targeting ER-β in TNBC could be an alternative approach for downregulating IGF/EGF signalling and controlling the impact of LDL in breast cancer patients.

## 1. Introduction

Breast cancer is the most common cancer worldwide in women and the leading cause of death, accounting for almost one in four cancer cases [1]. Obesity and high cholesterol levels have been established as having negative impacts on breast cancer mortality [2,3]. There are three main types of breast cancer, as determined by the types of receptors they possess: 70% of all breast cancer are hormone-receptor-positive, with either oestrogen (ER) or progesterone (PR) receptors; 15–20% are ERBB2 (HER2)-positive; and 15% have none of these receptors and are termed triple negative [4]. There are two main ERs, ER-β and ER-α, which have different transcriptional activity [4]. ER-α is expressed in 50–80% of breast cancers, whilst around 60% of all breast tumours that do not express ER-α are positive for ER-β expression [5]. Furthermore, nuclear ER-β and its variants are expressed in 44.4% of TNBCs [6]. Clinical studies have indicated that endocrine-targeted therapy for women with ER-α-positive breast cancer leads to significantly decreased mortality rates, by approximately 25–30% [7]. However, not all ER-α-positive breast cancer patients respond to endocrine therapy, and even those that are initially responsive ultimately become resistant as the disease progresses [7]. Additionally, some subtypes of TNBC are responsive to oestrogens [8]. ER-β has five different isoforms [6]. Different ER-β variants are expressed in TNBC; thus, the ER-β1 variant works as a tumour suppressor, while ER-β2 and β5 seem to act as pro-oncogenes in TNBC [8]. Furthermore, ER-β1 is more commonly expressed than the other ER-β2 and 5 isoforms. The roles of the ER-β variants remain unclear in TNBC. Recently, the role of ER-β in relation to the androgen receptor (AR) and its ability to mediate receptor-mediated effects were investigated, and it was suggested that the AR could be a marker in TNBC [8]. Studies have also found that the genomic actions of ER-β are regulated by ligand-modulated transcription factors that up- and/or downregulate gene expression in target tissues [9]. Taken together, oestrogen and its receptors play vital roles in breast cancer progression and treatment.

Altered cholesterol metabolism has emerged as an independent risk factor for breast cancer in post-menopausal women [2]. Mendelian randomisation, using genetic proxies constructed from genome-wide association studies, showed that high HDL and LDL levels were each causally associated with an increase in the risk of breast cancer [10]. The low density lipoprotein receptor (LDL-R) is abundant in TNBC cells and is important in their growth during the setting of elevated circulating LDL-C [11]. The mechanisms by which high cholesterol metabolism impacts on breast cancer risk and progression are still controversial.

Cholesterol molecules generate oxidized metabolites, termed cholesterol oxidation products (COP) or oxysterols [12]. The most abundant oxysterol in plasma is 27OHC, which is synthesised from cholesterol via the activation of the cytochrome P450 enzyme, sterol 27 hydroxylase, and cholesterol 27-hydroxylase (CYP27A1) [3]. Since researchers discovered that 27OHC was a selective endogenous modulator of ER-α, a number of studies have shown that the activation of ER-α by 27OHC in breast cancer cells culminates in increased cell proliferation and consequently promotes ER-α-positive breast cancer progression [13]. The proliferative role of 27OHC has been confirmed in vivo and in vitro using PyMT mice and murine human cancer cell xenografts [14,15]. In human breast cancer tissue, 27OHC concentrations and CYP27A1 protein expression are increased in higher grade tumours [14]. Furthermore, 27OHC had no effect on the proliferation of the ER-α-negative breast cancer cells SKBR3 and MDA-MB-231 [14]. The discovery of 27OHC as a selective endogenous modulator of ER-α that promotes ER-α-positive breast cancer proliferation could help to explain why some patients have resistance to aromatase inhibitors [16,17].

In lung cancer, 27OHC has also been shown to promote cell growth through ER-β and the activation of PI3K (phosphatidylinositol 3-kinase) and Akt (protein kinase B) signalling [18]. Previous studies have shown that 27OHC can bind to both ER-α and ER-β [19,20], though with greater affinity for ER-β [20].

Insulin-like growth factor (IGF) signalling and levels of cholesterol, LDL-C, and its oxysterol metabolite 27OHC are frequently upregulated in obese women [11]. Emerging evidence illustrates that metabolic factors, such as the IGF-I and IGF-II pathways, promote the progression of breast cancer [21,22]. Aberrant epidermal growth factor (EGF) and IGF signalling are detected in TNBC tumours and are linked to higher rates of recurrence, poor response to therapy, and reduced overall survival [23,24,25]. Studies have identified an association between ER-β and the IGF and EGF signalling cascades [26,27].

For this study, we established models in which we confirmed the effect of cholesterol and its metabolite, 27OHC, on breast cancer cell proliferation, migration, invasion, and the abundance of epithelial-to-mesenchymal (EMT) markers. Using these models, we determined novel associations between cholesterol metabolism with the IGF/EGF axes and their dependency on ER-α and/or -β.

## 2. Materials and Methods

27OHC (purity ≥ 99.9%) was purchased from Santa Cruz Biotechnology (Santa Cruz, CA, USA), dissolved in absolute ethanol to a stock solution of 1000 µM, and stored at −80 °C. A selective oestrogen receptor β (ER-β) antagonist (4-[2-Phenyl-5,7-is (trifluoromethyl)pyrazolo [1, 5-a]pyrimidin-3-yl]phenol (PHTPP)) was purchased from Sigma (Sigma-Aldrich, St. Louis, MO, USA), and low-density lipoprotein (LDL) from human plasma was bought from Thermo Fisher Scientific (Waltham, MA, USA).

### 2.1. Cell Culture

The human breast cancer cell lines MCF7 (epithelial) and MDA-MB-231 (mesenchymal) were obtained from ATCC (Molsheim, France). These cells were authenticated by short tandem repeat (STR) analysis and are confirmed as mycoplasma-negative in our routine quality control. All cell lines were cultured as described previously [28].

### 2.2. SiRNA Transfection

Transfection was performed using a nontargeting negative control siRNA (Dharmacon, Chicago, IL, USA) or siRNA to ER-α (L-003401-00-0020; SMARTpool: ON-TARGETplus human siRNA ESR1; Dharmacon, GE Healthcare, Chicago, IL, USA), ER-β targeting oestrogen receptor 2 (ESR2), (SMARTpool: ON-TARGETplus human siRNA ESR2, L-003402-00-0020; Dharmacon, GE Healthcare, Chicago, IL, USA), and CYP27A1 (cytochrome P450 family 27 subfamily A member 1) (siRNA ID s3887) obtained from Thermo Fisher Scientific (Waltham, MA, USA). Downregulation was achieved via transfection using Saint-red (Synvolux, Leiden, The Netherland) according to the manufacturer’s protocol. Lyophilised siRNA was resuspended with an appropriate amount of an siRNA suspension buffer to make a 20 μM stock solution, which was aliquoted and stored at −20 °C. The concentration of the siRNA in the transfection master-mix depended on its required final concentration. The stock cells were trypsinised when 70–80% confluent, counted, and then seeded into 6-well plates (0.1 × 10^6^/cells in 800 µL/well of GM). Then, 200 µL of the transfection master-mix were gently added dropwise. The cells were incubated for 24 h before changing the media to SFM for a further 24 h at 37 °C in 5% CO_2_. The transfection efficiency was assessed by Western immunoblotting.

### 2.3. Cell Viability

This was determined using a crystal violet (CV) proliferation assay [29]. In brief, cells were seeded into 96 well plates (Greiner bio-one, cat #655090) at a density of 5000 cells/well. After incubation, the cells were fixed by removing the medium and adding 4% paraformaldehyde (PFA) for at least 20 min. Then, the cells were washed twice with phosphate buffered saline followed by the addition of 0.05% crystal violet solution (100 μL). The plate was then washed and 1% sodium dodecyl sulphate (SDS); SDS /PBS was added to solubilize the CV and the optical densities of the extracts were measured at 590 nm using a plate reader (FLUOstar Optima, BMG LABTECH, Ortenberg, Germany). Cell proliferation for siRNA transfection experiments was measured using the Muse™ cell analyser (Millipore, Burlington, MA, USA).

### 2.4. Western Blotting

Western blot analysis was performed as described previously [28]. In brief, protein cell lysates (30 μg), were subjected to SDS polyacrylamide gel electrophoresis, transferred to a nitrocellulose membrane (BioRad, Watford, UK), and immunoblotted with the following antibodies: anti-CYP27A1 (1:1000, ab227248; Abcam, Cambridge, UK), anti-liver-X-receptor-β (LXR-β) (1;1000, D6M9D; Cell Signalling Technology, Danvers, MA, USA), fibronectin (1:500, 611,447; BD Biosciences, Franklin Lakes, NJ, USA), E-cadherin (1:1000, 610,181; Cell Signalling), vimentin (1:500, 550513; BD Biosciences), ER-β (1:1000, PPZ0506; Thermo Fisher, Waltham, MA, USA), GAPDH (1:5000, CB1001; Millipore, Burlington, MA, USA), β-actin (1:10,000, A1978; Sigma-Aldrich, St. Louis, MO, USA), and ER-α (1:1000, sc-8002; Santa Cruz, Santa Cruz, CA, USA). After incubation with secondary antibodies conjugated to peroxidase (Sigma, St. Louis, MO, USA), proteins were detected with a Clarity ECL substrate (BioRad, Watford, UK) using BioRad Chemidoc XRS + system and quantified using Image software.

### 2.5. Trans-Well Cell Migration/Invasion Assay

At 48 h post-treatment, 1.5 × 10^5^ cells were seeded in triplicate into 8 μM pore trans-well inserts (MCEP24H48; Millipore, Burlington, MA, USA) coated with collagen for measuring invasion or without collagen for assessing migration. DMEM with 5% foetal bovine serum, which was used as a chemoattractant solution, was placed in the lower chamber towards which the cells migrated for 24 h (MCF-7) or 6 h (MDA-MB-231) at 37 °C. The incubation times used to assess cell migration and invasion reflected the invasive potential and aggressiveness of the cells. The cells in the filter were fixed using 4% PFA, and the cells in the upper compartment of the insert were removed by gently wiping the upper side of the membrane with a cotton swab. Then, cells on the underside were fixed, stained with 0.05% CV, and permeabilised. The stain was then dissolved in 1% SDS and placed the plate on shaker for one hour before we read the optical density values using iMark plate reader (BioRad, Watford, UK) at 595 nm.

### 2.6. Molecular Taxonomy of Breast Cancer International Consortium (METABRIC) and the Cancer Genome Atlas (TCGA) Dataset Analysis

The TCGA and METABRIC data accessed during this study are available in a public repository from the cBioPortal for Cancer Genomics website (http://www.cbioportal.org/ accessed on 11 July 2021). TCGA and METABRIC microarray expression data from 1082 and 1904, respectively, invasive breast cancer carcinomas were derived from the cBioPortal (dataset described in [30]). Data from patients whose gene expression for CYP27A1 or ER-β read as ‘‘Not a Number’’ were discounted, which resulted in 504 usable samples for TCGA and 1193 for METABRIC. Array gene expression data were analysed based on the median of CYP27A1 expression.

### 2.7. Radioimmunoassay (RIA)

The concentrations of IGF-I peptide in MDA-MB-231 cell supernatants were measured using an ‘in-house’ RIA, as previously described [31].

### 2.8. Statistical Analysis

Experiments were repeated three times in triplicate. GraphPad Prism 8.0.1 software for Mac (La Jolla, CA, USA) was used to determine the mean ± standard error of the mean (SEM). To analyse the data, one-way ANOVA was used followed by the least significant difference (LSD) post-hoc test. TCGA and METABRIC microarray data were tested for normal distribution followed by a non-parametric Mann–Whitney test for significance.

## 3. Results

### 3.1. Cholesterol LDL and Its Metabolite 27OHC Increases Cells Proliferation, Migration, and Levels of EMT Markers in Breast Cancer Cells

We firstly confirmed the effects of LDL (0–100 μg/mL) on cell proliferation in an ER-α-positive cell line, MCF-7, and established that it also had similar effects on the proliferation of an ER-α-negative breast cancer cell line, MDA-MB-231.

Cell proliferation was increased with both doses of LDL by 76.4% at 80 μM and 58.9% at 100 μM in MCF-7 cells (Figure 1A) and by 72.8% at 80 μM and 77.3% at 100 μM in MDA-MB-231 cells (Figure 1B) compared to the control. Though 27OHC promoted MCF-7 cell growth at 0.1 µM by 48.2% (Figure 1C), there was no effect of 27OHC on the growth of MDA-MB-231 cells (Figure 1D) relative to the control.

Cell migration was increased with LDL and 27OHC treatment in MCF-7 (Figure 1E) and MDA-MB-231 cells (Figure 1F) compared to the controls. Cell invasion was significantly increased with LDL and 27OHC in MCF-7 (Figure 1G) and MDA-MB-231 cells (Figure 1H) compared to controls. These data suggest that LDL and 27OHC promote MCF-7 and MDA-MB-231 cell invasion and migration.

To assess the effects of 27OHC and LDL on the abundance of EMT markers, we used both MCF-7 (epithelial) and MDA-MB-231 (mesenchymal) breast epithelial cells. The Western blots results show the effect of 27OHC and LDL on the abundance of EMT markers (Figure 1I,K, respectively) in comparison to the controls. With MCF-7 cells, we observed a reduction in the levels of E-cadherin with 27OHC and LDL and an increase in fibronectin. With MDA-MB-231 cells, we found increases in both the abundance of fibronectin and vimentin levels when exposed to 27OHC and LDL, respectively.

### 3.2. Cholesterol Increases Proliferation and Migration through 27OHC Production in Breast Cancer Cells

We next confirmed that the effects of LDL on proliferation were mediated by CYP27A1 (the enzyme responsible for the rate-limiting step in 27-hydroxycholesterol biosynthesis). With MCF-7 cells, LDL increased proliferation and migration, but with CYP27A1 silenced, these effects were blocked (Figure 2A,B).

Similarly, with MDA-MB-231 cells, LDL promoted cell growth and migration, and these effects were blocked with CYP27A1 silenced (Figure 2D,E). The Western blot results demonstrate the effective silencing of CYP27A1 in both MCF7 (Figure 2C) and MDA-MB-231 cells (Figure 2F).

### 3.3. 27OHC Promotes Cell Proliferation, Migration, and Invasion of Breast Cancer Cells

Silencing ER-α alone significantly reduced the proliferation of MCF-7 cells (Figure 3A,B) but had no effect on MCF-7 cell migration (Figure 3C) or invasion (Figure 3D). 27OHC promoted the growth, migration, and invasion of MCF-7 cells (Figure 3A,C,D, respectively), whilst the effects of 27OHC on cell growth were blocked with ER-α silenced (Figure 3A) and its actions on migration and invasion were unaffected (Figure 3A,C,D, respectively). These results demonstrate that ER-α only mediated the effects of 27OHC on cell proliferation, not its impact on cell migration and invasion.

To examine the mechanism by which 27OHC was promoting cell migration and invasion, we silenced the ER-α in the ER-α-positive cells and assessed the abundance of alternative receptors to which 27OHC could bind: LXR-β and ER-β. Our data suggest that silencing ER-α only increased the abundance of ER-β in the ER-α-positive cell line while reducing LXR-β (Figure 3E). This suggested that LXR-β was not involved but ER-β may play a role in the effects of 27OHC on migration and invasion.

### 3.4. 27OHC Increases Cell Migration and Invasion of MDA-MB-231 Breast Cancer Cells

To assess whether ER-β played a role in the actions of 27OHC on breast cancer cell migration and invasion, we chose the MDA-MB-231 cells because they possess ER-β but lack ER-α.

27OHC increased cell migration and invasion, and these effects were blocked with ER-β silenced (Figure 4A–C, respectively). We confirmed these observations by using a selective ER-β antagonist, PHTPP: 27OHC was similarly unable to increase migration in the presence of PHTPP (Figure 4D). Notably, using the publicly available METABRIC and TCGA gene expression databases, we found that tumours that had high mRNA levels of CYP27A1 (and thus expected elevated 27OHC) also presented the increased expression of ER-β mRNA. Conversely, the low tumour expression of CYP27A1 was associated with the lower expression of ER-β (Figure 4E).

### 3.5. The Involvement of the IGF System in the Actions of Cholesterolin TNBC

LDL treatment increased the cell proliferation of MDA-MB-231 cells in comparison to the controls, whereas with the IGF-IR antagonist AG1024 present, LDL was no longer able to increase cell growth (Figure 5A) We had previously shown that this dose of AG1024 was effective in that it blocked the IGF-I-induced phosphorylation of IGF-IR (data not shown). The ability of LDL to increase MDA-MB-231 migration was also significantly inhibited in the presence of AG1024 (Figure 5B). Treatment with 27OHC (0.1 µM) and LDL (80 µg/mL) for 48 h increased the abundance of IGF-IR in MDA-MB-231 (Figure 5C) in comparison to the controls, and this was accompanied by an increase in the production of IGF-I by MDA-MB-231 cells (Figure 5D), thus suggesting a positive feedback loop in response to cholesterol and 27OHC. In addition, the effects of LDL on increasing IGF-I secretion were mediated by 27OHC, as this action of LDL was inhibited with CYP27A1 silenced (Figure 5E). We showed that LDL also increased the proliferation of MCF-7 cells (Figure 2) and that this was similarly associated with increased levels, the activation of IGF-1R, and the phosphorylation of AKT (Appendix A).

Data from the publicly available METABRIC and TCGA gene expression databases indicated that tumours with high levels of CYP27A1 (and thus expected elevated 27OHC) also presented an increased expression of IGF-I. Conversely, the low tumour expression of CYP27A11 was associated with the lower expression of IGF-I (Figure 5F,G).

### 3.6. ER-β Regulates IGF-1 and EGF Receptors in TNBC

Treatment with a synthetic ER-β agonist increased levels of the IGF-I receptor (from 1 µM with a peak response observed at 2.5 µM) and the EGF receptor (EGFR) at 2.5 µM in comparison to the controls; Figure 6A,B. Conversely, with ER-β inhibition, using PHTPP, there was a reduction in the levels of IGF-IR at 5 and 10 µM and the levels of EGFR at 10 µM (Figure 6C,D). Furthermore, PHTPP caused the inhibition of cell proliferation in MDA-MB-231 cells from 34.4% at 2 µM to 80.8% at 10 µM (Figure 6E). Interestingly, using the publicly available METABRIC and TCGA gene expression databases, tumours that had high mRNA levels of ER-β also presented the increased expression of IGF-IR, EGFR, and Ki-67, a marker of proliferating cells. Conversely, the low tumour expression of ER-β was associated with the lower expression of IGF-IR, EGFR, and Ki-67 (Figure 6F).

## 4. Discussion

Several studies have indicated a role for cholesterol metabolites in enhancing tumour growth and invasion [3,17,19,32]. Cholesterol-lowering drugs (such as statins) have been shown to have protective effects against breast cancer deaths and recurrence [33], suggesting that lowering cholesterol and 27OHC may have therapeutic benefits against breast cancer development. Nevertheless, the precise mechanism of how cholesterol affects breast cancer pathogenesis, especially regarding ER-β, is still unclear [34].

We found that LDL drives the proliferation, migration, and invasion, as well as increasing the abundance of, mesenchymal markers of ER-α-positive and ER-α-negative breast cancer cells, which was consistent with a previous study examining the importance of LDL-R in the growth of TNBCs [11].

LDL-induced growth in both ER-α-positive and ER-α-negative cells was inhibited when CYP27A1, the enzyme responsible for the rate-limiting step in 27OHC biosynthesis, was silenced using siRNA, suggesting that 27OHC could contribute to the risk of breast cancer by mediating LDL-induced breast cancer proliferation. These observations are consistent with previous findings [14,15]. The same results have also been observed in lung cancer cells, where exogenous 27OHC supplementation was found to increase cell proliferation [18], suggesting that 27OHC signalling mediated the effects of cholesterol.

Though LDL-induced proliferation in ER-α-negative cells was blocked when CYP27A1 was silenced, we surprisingly found that 27OHC alone was unable to increase cell growth in these cells. These data may suggest the possibility that other metabolites of cholesterol, such as 25-hydroxycholesterol (25OHC), may also be involved. 25OHC is catalysed from cholesterol via different enzymes such as CH25H, cytochrome P450 3A4, (CYP3A4) and CYP27A1, and it can be catalysed by free radical oxidation [35]. This metabolite has been shown to promote lung cancer cell migration and invasion [36]. Therefore, the potential role of 25OHC in mediating some of the actions of cholesterol warrants further investigation.

We showed that 27OHC promoted cell growth in ER-α-positive breast cancer cells, which had been previously observed [3,19,20]. Silencing ER-α blocked the proliferative effect of 27OHC, in line with evidence suggesting a role for cholesterol metabolites in promoting the growth of ER-α-positive breast cancers through acting as endogenous SERMS [3,19,20]. We previously showed in MCF-7 cells that the activation of ER-α resulted in its phosphorylation and translocation to the nucleus, where it upregulated the expression of target genes [37].

Notably, 27OHC was also able to promote breast cancer cell migration and invasion in both cell lines, and silencing ER-α did not affect this in ER-α-positive cells but was accompanied by an increase in the abundance of ER-β.

The role 27OHC plays has been shown to depend on the relative ratio of the expression of ER-α to ER-β, with ER-β having a protective effect in the presence of ER-α; however, recent data suggest that in the absence of ER-α, ER-β has a pro-tumorigenic role [38,39,40]. A recent study indicated that in breast cancer cell lines, ER-β inhibited migration through CLDN-6-mediated autophagy [41].

Our data showing the link between 27OHC and ER-β are consistent with the results from the TCGA and METABRIC database, as we found that tumours that had high mRNA levels of CYP27A1 (and thus expected elevated 27OHC) also presented the increased expression of ER-β, suggesting that 27OHC may be a positive regulator of ER-β expression in breast cancer tumours [42].

We found that 27OHC-induced cell migration and invasion through ER-β, suggesting that 27OHC can act via both ER-α or ER-β in our models depending on the context. This is supported by the observation that 27OHC increases cell proliferation in prostate cancer cell lines via ER-β activation [43]. In contrast, it has also been reported that 27OHC acts as a negative modulator of ER-β in Ishikawa cells [44]. Since 27OHC shows both agonistic and antagonistic effects on ER-α function depending on target organs [45], 27OHC may also function as both an antagonist and agonist for ER-β depending on the particular tissue/cells or the presence or absence of ER-α [5] A recent study suggested an association between ER-β and AR, where ER-β and AR act together in TNBC to modulate cell function [8]. Further work would be required to confirm the relative contribution of AR to the actions of 27OHC on breast cancer cells.

In overweight women, it has been shown that IGF signalling and levels of cholesterol, specifically LDL-C, are upregulated. It was recently found that LDL-R abundance is important in the growth of TNBCs in the setting of increased circulating LDL-C and is thought to be a major contributing factor to the increased mortality and recurrence in overweight women with TNBC [11]. We found that the addition of LDL increased the proliferation and migration of MDA-MB-231 TNBC cells, which was accompanied by an increase in the production of IGF-I and the abundance of the IGF-IR. These effects of LDL were blocked in the presence of an IGF-1R tyrosine kinase inhibitor, suggesting a positive feedback loop in response to cholesterol and 27OHC that we similarly observed in MCF-7 cells. An autocrine loop has also been described for other growth factors, such as the nerve growth factor (NGF) that modulates an autocrine loop through the activation of tyrosine kinases, that may contribute to the sustained growth and aggressiveness of TNBC [46]. It would be interesting in the future to examine these effects in more physiologically relevant models, including colony formation and growing cells in 3D.

This phenotype has been confirmed in prostate cancer cells where the statin simvastatin deregulated IGF-IR expression [47] and impeded both basal IGF-I and IGF-I-induced ERK and Akt activation [48].

Simvastatin has also been proposed to inhibit IGF-IR in bile duct cancer cells [49], suppress oesophageal cancer cell growth, and lead to the development of radio-resistance by reversing the process of the epithelial-to-mesenchymal (EMT) transition through the PTEN-PI3K/AKT signalling pathway. These phenotypic effects were blocked by the addition of IGF-I [50,51]. In vitro experiments in human hepatocytes cells demonstrated a role for the growth hormone IGF-I and glucocorticoids in the regulation of the activity of the enzyme CYP27A1 [52]. Notably, we determined that the ability of LDL to increase the secretion of IGF-I and its receptor was inhibited when the enzyme CYP27A1 was silenced in TNBC. These data are consistent with the results from the TCGA and METABRIC databases, which showed a positive association between levels of IGF-IR with CYP27A1. These phenotypic effects that we observed have been reported by others with different methods of targeting cholesterol metabolism [50,53].

Functional crosstalk exists between ERs and growth factor receptors, such as EGFR and IGF-IR. Growth factor pathways can activate oestrogen receptors, and oestrogen receptors can activate IGF signalling [54]. The treatment of ER-α-positive breast cancer cells with tamoxifen can lead to resistance, and IGF-IR and/or EGFR are critical for breast cancer resistance to endocrine therapy [55]. The role of ER-β in breast cancer growth has been shown in ER-α-negative TN tumours [56], lung cancer [57], and prostate cancer [44]. There have been conflicting reports about the role of ER-β in breast cancer [27,58,59,60], due to differences between studies; this could be related to the lack of standardized detection methods and insufficiently validated antibodies [43,61,62]. Moreover, ER-β has five different isoforms and variants, which could also complicate the investigation of the physiological role of ER-β and its participation in the carcinogenesis of breast cancer [6,60,63]. It has been shown that ER-β1, ER-β2, and ER-β5 are expressed in breast cancer tissue [63,64]. To address these differences, we used a certified TNBC cell line (negative for ER-α, PR, and HER2) and validated ER-β antibodies [64].

With ER-β inhibition, we observed suppression of IGF-I and EGF receptors and an inhibition of basal cell proliferation. Similar findings were recently reported by others in TNBC, but they observed an interaction between ER-β and insulin-like growth factor-II (IGF-II) [27,62]. They also indicated, by silencing ER-β in TNBC cell lines, that reduced cell proliferation, migration, and invasion were associated with the downregulation of the IGF-I and EGF receptors. Conversely, upregulating ER-β using an agonist increased levels of IGF-1R [26,27]. We further determined that upregulating ER-β using an agonist increased levels of the IGF-I and EGF receptors in TNBC. These data suggest that ER-β is a positive regulator of the IGF-I and EGF receptors. This is consistent with the results from the TCGA and METABRIC databases, in which we found an association between the levels of [48] ER-β with the levels of IGF-IR, EGFR, and Ki67 mRNA. These data imply that ER-β is a positive regulator of IGF-I and EGF receptors and proliferation.

A link between ER-β with the IGF and EGF signalling pathways has also been previously reported, where silencing ER-β in TNBC cell lines reduced cell proliferation, migration, and invasion, which were associated with the downregulation of the IGF-I and EGF receptors [26,27]. These biological links were also observed at the genomic levels, as illustrated with data from the TCGA and METABRIC datasets.

Different strategies have been developed to target the IGF axis based on convincing pre-clinical data that suggest a role for IGF signalling pathways in cancer. The use of monoclonal antibodies and small molecules targeting the tyrosine kinase domain of IGF-IR has comprised the major approaches to disrupting IGF signalling cascades [25]. However, due to a lack of a strategy for selecting responsive patients, these strategies applied as monotherapies have had limited clinical success. Some studies have indicated that ER-β is a biomarker linked to more aggressive breast cancer [65], and our data may suggest that ER-β may be biomarker of response to anti-IGF-IR therapy or that targeting ER-β in TNBC could be a different approach for downregulating IGF signalling. Further work would be required to confirm a functional role of EGF signalling similar to that of IGF-IR.

## 5. Conclusions

In summary, our study elaborates a mechanism to support clinical studies suggesting a link between obesity and high cholesterol with an increased risk of breast cancer progression: cholesterol promotes cell proliferation, migration, and invasion, and we have presented novel data that indicate that ER-β is central to the effects of LDL/27OHC on invasion and migration. We suggest that there is a positive feedback loop in response to cholesterol and its metabolites in breast cancer cells, and this loop involves increased tyrosine kinase receptor phosphorylation and ER activation, with subsequent nuclear translocation, where they upregulate target genes (including IGF-I, IGF-IR, and EGFR). Delineating this suggested signalling pathway may identify novel opportunities for optimizing current breast cancer treatment regimens.

## Figures and Tables

**Figure 1 cells-11-00094-f001:**
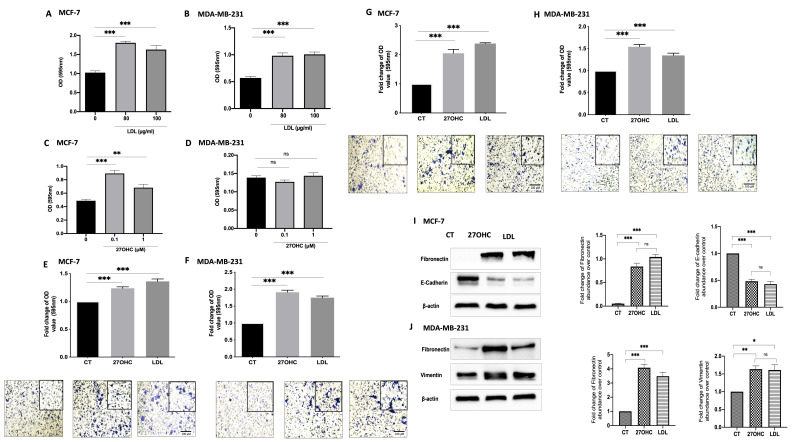
Effects of exogenous 27OHC and LDL on cell growth, migration/invasion, and EMT markers. Crystal violet staining proliferation assay results for (**A**) MCF-7 and (**B**) MDA-MB-231 after being dosed with LDL (80 and 100 μg/mL) for 48 h and with 27OHC (0.1 and 1 μM) for 48 h for (**C**) MCF-7 and (**D**) MDA-MB-231. A trans-well assay was used to detect cell migration and invasion after being dosed with LDL or 27OHC, the migrated cells were stained with crystal violet, and images were taken (×20 magnification). Quantification of (**E**) MCF-7 cell migration after a 24 h incubation period, (**F**) MDA-MB-231 cell migration after 6 h incubation period. Quantification of (**G**) MCF-7 cell invasion after 24 h incubation period and (**H**) MDA-MB-231 cell invasion after a 6 h incubation period. Western immunoblot analysis was performed to show the protein abundance of EMT markers, fibronectin, and vimentin, and their densitometric analyses were corrected in (**I**) MCF-7 and (**J**) MDA-MB-231 after being dosed with 27OHC (0.1 μM) and LDL (80 μg/mL). *p*-values were determined by using one-way statistical analysis in GraphPad Prism: ANOVA test plus least significant difference (LSD) post-hoc test (* *p* < 0.05, ** *p* < 0.01, *** *p* < 0.001). Scale bar represents 100 μM.

**Figure 2 cells-11-00094-f002:**
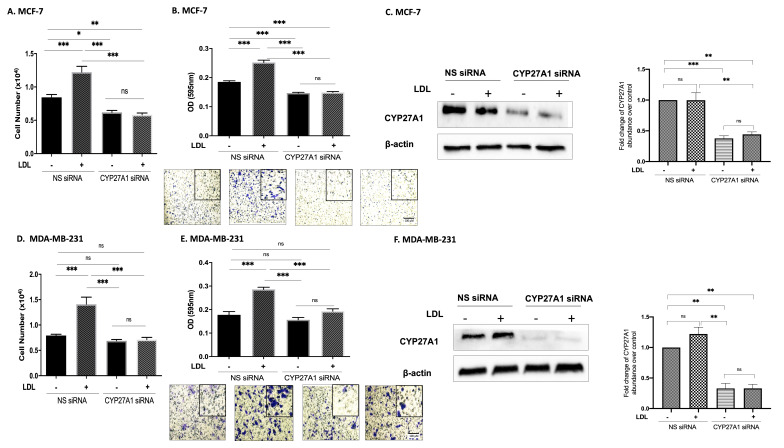
Cholesterol increases proliferation and migration through 27OHC production in breast cancer cells. Using the Muse cell analyser, we assessed the cell number for (**A**) MCF-7 and (**D**) MDA-MB-231 after being transfected with CYP27A1 siRNA and non-silencing RNA and being dosed with (LDL 80 μg/mL) for 48 h. A trans-well assay was used to detect cell migration and invasion after being transfected with CYP27A1 siRNA and dosed with (LDL 80 μg/mL). The migrated cells were stained with crystal violet, and images were taken (×20 magnification). The migrated cells were quantified for (**B**) MCF-7 after 24 h and (**E**) MDA-MB-231 after 6 h of incubation time. Western immunoblot analysis was performed to show the protein abundance of CYP27A1 MCF (**C**) for MCF-7 and (**F**) MDA-MB-231 with or without LDL and CYP27A1 silencing. Additionally, the relative fold changes of CYP27A1 against loading control β-actin were measured (**C**,**F**). *p*-values were determined by using the one-way statistical analysis of GraphPad Prism: ANOVA test plus least significant difference (LSD) post-hoc test (* *p* < 0.05, ** *p* < 0.01, *** *p* < 0.001). Scale bar represents 100 μM.

**Figure 3 cells-11-00094-f003:**
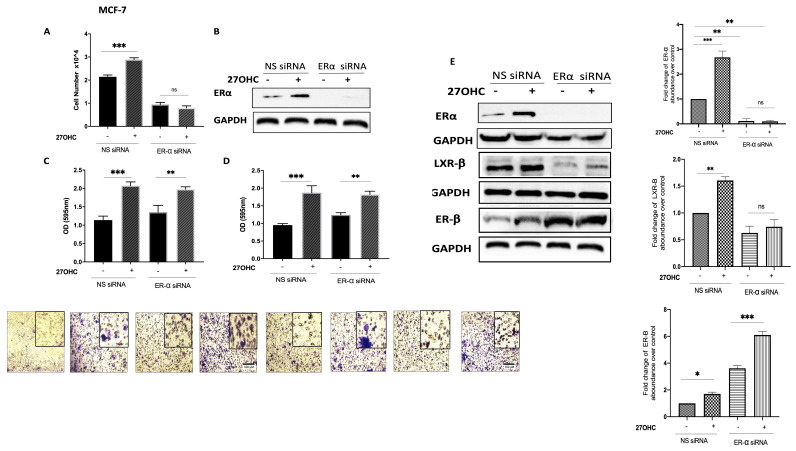
Effects of 27-hydroxycholesterol on cell growth and migration/invasion in the presence or absence of ER-α. Using the Muse cell analyser, we assessed the cell number for (**A**) MCF-7 after being transfected with ER-α siRNA (20 nm) and non-silencing RNA (20 nm) and dosed with 27OHC (0.1 μM) for 48 h. (**B**) Western immunoblot analysis was performed to show the protein abundance of ER-α in MCF-7 with or without ER-α silencing and 27OHC. β-Actin was used as a loading control. A trans-well migration/invasion assay was used for the migrated cells after 24 h, the cells were stained with crystal violet, and images were taken (×20 magnification). We quantified (**C**) cell migration and (**D**) invasion in MCF-7 cells after being transfected with ER-α siRNA and dosed with 27OHC. (**E**) Densitometry of the Western blot showing the protein abundance of ER-α, LXR-β, and ER-β after normalization to the reference protein GAPDH and B-actin. *p*-values were determined by using the one-way statistical analysis of GraphPad Prism: ANOVA test plus least significant difference (LSD) post-hoc test (* *p* < 0.05, ** *p* < 0.01, *** *p* < 0.001). Scale bar represents 100 μM.

**Figure 4 cells-11-00094-f004:**
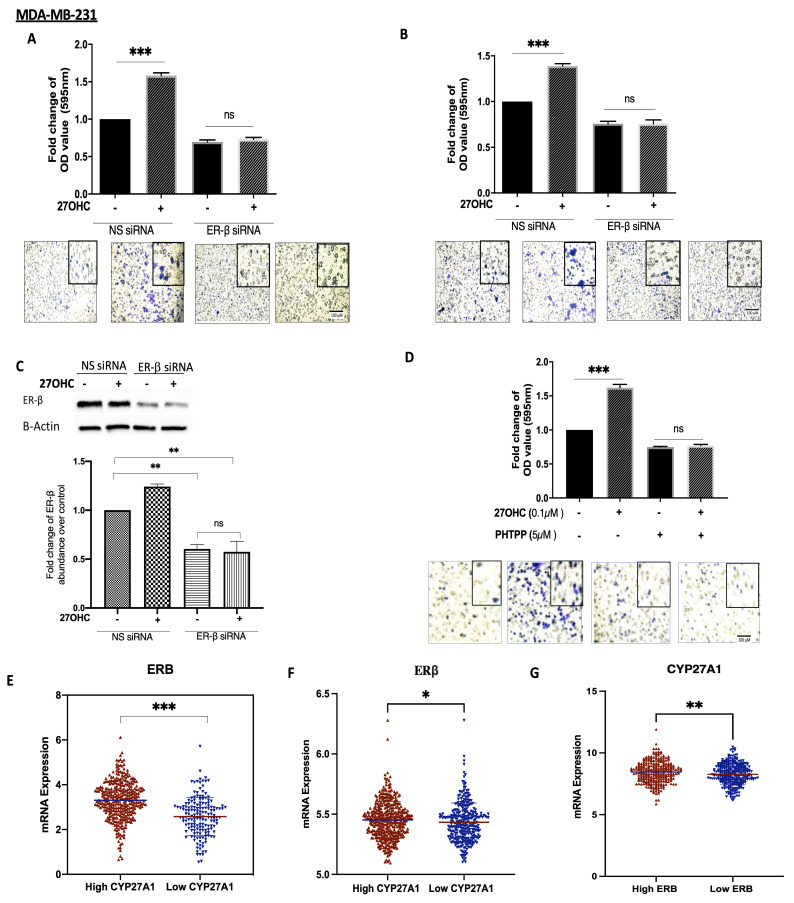
27OHC increases cell migration/invasion in ER-α-negative breast cancer cells via ER-β. A trans-well assay was used to detect (**A**) migration and (**B**) invasion after 6 h for MDA-MB-231 after being transfected with ER-β siRNA (60 nm) and non-silencing RNA (60 nm) and dosed with 27OHC (0.1 μM). (**C**) Western blotting was used to detect the abundance of ER-β with or without with ER-β silencing. The densitometry quantification of ER-β was assessed after normalization to β-actin. (**D**) We quantified cell migration after being dosed with 27OHC (0.1 μM) and PHTPP (5 μM). The migrated and invaded cells were stained with crystal violet, and images were taken (×20 magnification). (**E**) Scatterplot analysis of the TCGA data of 504 invasive breast cancer carcinomas for the mRNA expression of ER-β represented as the median expression of CYP27A1. (**F**) Scatterplot analysis of the METABRIC data of 1193 invasive breast cancer carcinomas for the mRNA expression of IGF-I represented as median expression of CYP27A1. (**G**) Scatterplot analysis of the TCGA data of 504 invasive breast cancer carcinomas for the mRNA expression of CYP27A1 represented as the median expression of ER-β. Results are presented as mean +/− SEM in (**A**–**D**); in (**E**), the horizontal line presents the mean. *p*-values were determined by using the one-way statistical analysis of GraphPad Prism: one-way ANOVA followed by least significant difference (LSD) post-hoc test (* *p*  <  0.05, ** *p*  <  0.01, *** *p*  <  0.001) (**A**–**D**); (**E**) Mann–Whitney test. Scale bar represents 100 μM.

**Figure 5 cells-11-00094-f005:**
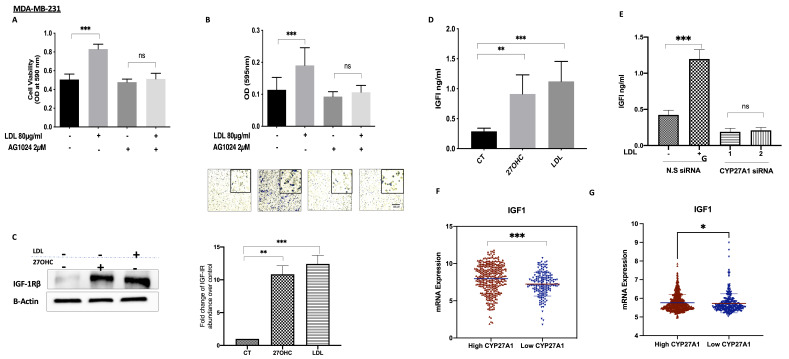
Interactions between the IGF system and cholesterol metabolism in TNBC. Cell proliferation for (**A**) MDA-MB-231 treated with (LDL 80 μg/mL) in the presence or absence of a tyrosine kinase inhibitor (AG1024; 2 µM) for 48 h, assessed using a crystal violet proliferation assay. Cell migration for (**B**) MDA-MB-231 cells after being dosed with LDL with or without AG1024 (2 µM). The migrated cells were stained with crystal violet, and images were taken (×20 magnification). Western blotting results for (**C**) MDA-MB-231 after being dosed with 27OHC (0.1 µM) and LDL (80 µg/mL) for 48 h; their densitometry analyses of fold changes of IGF-IRβ against loading control β-actin. A radioimmunoassay was used to measure IGF-I concentrations in (**D**) MDA-MB-231 cells after being dosed with 27OHC (0.1 µM) and LDL (80 µg/mL) for 48 h, as well as (**E**) levels of IGF1 in the presence or absence of CYP27A1 in MDA-MB-231 after being dosed with LDL in the presence or absence of CYP27A1 for 48 h. (**F**) Scatterplot analysis of the TCGA data of 504 invasive breast cancer carcinomas for the mRNA expression of IGF-I represented as the median expression of CYP27A1. (**G**) Scatterplot analysis of the METABRIC data of 1193 invasive breast cancer carcinomas for the mRNA expression of IGF-I represented as the median expression of CYP27A1. Data representative of mean ± SEM (*n* = 3). *p*-values were determined by using the one-way statistical analysis of GraphPad Prism: ANOVA test plus least significant difference (LSD) post-hoc test (* *p* < 0.05, ** *p* < 0.01, *** *p*< 0.001) (**A**–**F**); (**G**) Mann–Whitney test. Scale bar represents 100 μM.

**Figure 6 cells-11-00094-f006:**
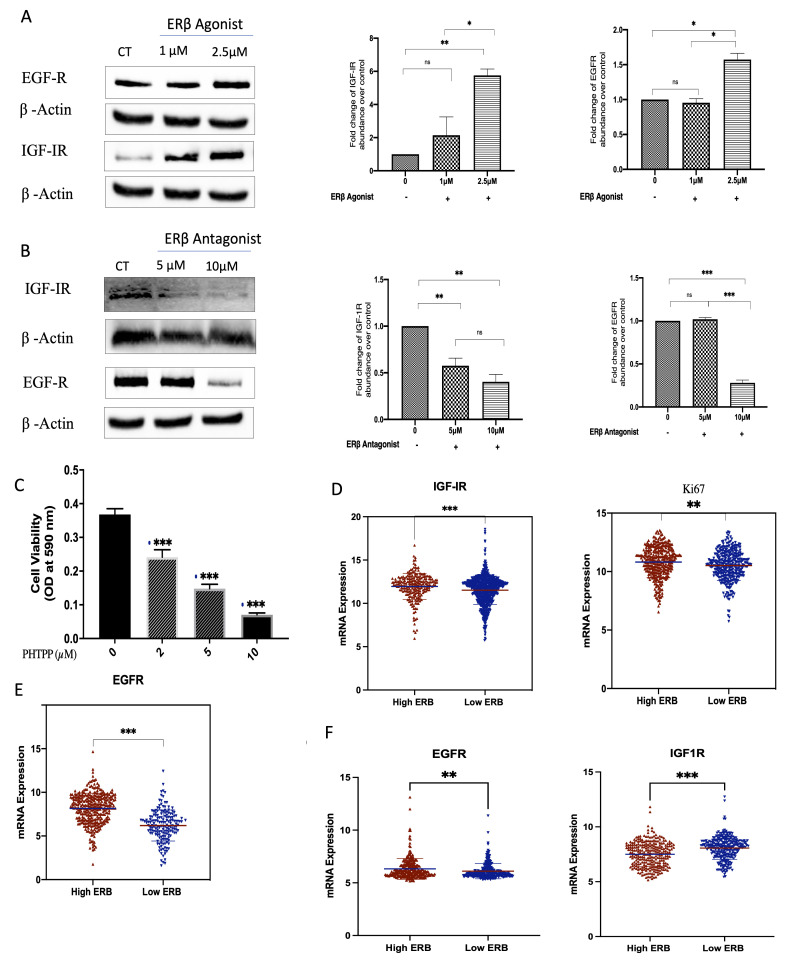
Effects of selective ER-β agonist and antagonist treatment on levels of IGF-IR and EGFR in MDA-MB-231. MDA-MB-231 were treated with an ER-β agonist or antagonist (**A**,**B**) for 24 h and were immunoblotted for IGF-IR and EGFR. GAPDH was used as a loading control, and respective quantitative densitometry analyses of the blots are shown in MDA-MB-231 cells. (**C**) MDA-MB-231 cells were treated with increasing doses of a selective ER-β antagonist, PHTPP, for 48 h, and cell proliferation was assessed using a crystal violet assay. (**D**,**E**) Scatterplot analysis of the TCGA data of 504 invasive breast cancer carcinomas for the mRNA expression of IGF-IR, Ki67 (**D**), and EGFR (**E**) represented as the median expression of ER-β. (**F**) Scatterplot analysis of the METABRIC data of 1172 invasive breast cancer carcinomas for the mRNA expression of EGFR and IGF-1R represented as the median expression of ER-β. Results are presented as mean +/− SEM (*n* = 3) (**A**–**C**), where the horizontal line presents the mean. *p*-values were determined by using the one-way statistical analysis of GraphPad Prism: (**A**–**C**) one-way ANOVA test followed by least significant difference (LSD) post-hoc test (* *p* < 0.05, ** *p* < 0.01, *** *p* < 0.001); (**D**,**E**) Mann–Whitney test.

## Data Availability

The TCGA and METABRIC data accessed during this study are available in a public repository from the cBioPortal for Cancer Genomics website (http://www.cbioportal.org/ accessed on 11 July 2021).

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
