# Peer review of "A Role for ER-Beta in the Effects of Low-Density Lipoprotein Cholesterol and 27-Hydroxycholesterol on Breast Cancer Progression: Involvement of the IGF Signalling Pathway?"

_cells, 2021, doi:10.3390/cells11010094_

Round 1

Reviewer 1 Report

In this manuscript by Mashat et al., the authors explored the effect of cholesterol and its metabolite 27 OHC on breast cancer cell proliferation, migration, and invasion. They also explored the association of the IGF and EGF axes with cholesterol metabolism and particularly if these effects involved the estrogen receptor beta.

This referee strongly believes that major modifications should be done to the manuscript in order to improve its clarity to the reader.

1-In the Introduction the authors indicate (line 38): “Tumours expressing progesterone receptor (PR), estrogen receptor (ER) and/or the epidermal growth factor receptor 2 (HER2) are defined as hormone receptor-positive breast cancers, and triple-negative breast cancers (TNBC) lack PR, ER or HER2 expression [4]”. This referee believes that a more accurate definition of HR-positive BC should be included in the manuscript. Indeed, HR-positive breast cancers are the ones displaying ER and/or PR, independently of HER2 overexpression. There is a group of HR+ BCs that is HER-2 negative (HR+/HER2-: Luminal A- and Luminal B-like) and another group of HR+ BCs that are HER2-positive. On the other hand, HER2-positive BCs could be either HR-positive or HR-negative.

2- This reviewer found that the English language, grammar, and style of the Introduction and Discussion sections are adequate and engaging. However, a detailed revision of the English language and style is required in the Results section. Particularly, there are some words that are not usually used in scientific articles such as the word “negated”, whose meaning is not clear to this referee. Besides, this reviewer recommends the authors include a brief description of the aim and the BC models used in the first section of results (section 3.1). In addition, the results section would benefit from titles that would better highlight the key findings of each section.

3- This reviewer suggests that some modifications in the figure's style and figure legends would be friendlier to the readers. It would be interesting to include the cell line name in each panel. Please include the WB images and their densitometric analysis in the same panel (Figure 1 panels I-L; Figure 5 panels C and D; Figure 6 panels A-D). Regarding the images corresponding to invasion assays (Figure 1 panels E to H;  Figure 2 panels B and E; Figure 3 panels C and D; Figure 4 panels A-C; Figure 5 panel B), please clearly indicate at which region from the image at 20X the inset corresponds. Please note that the statistic symbols are missing in: Figure 1 panels A, B, C, E, F, G, H, J, and L; Figure 2 all panels;  Figure 3 panels A, C, D, and E; Figure 4 panels A-D; Figure 5 all panels. Regarding Figure Legends, please refer to the conventional style of description of the panels in which each panel ID is indicated before the description of the panel and also check that the panels are described consecutively as they are mentioned and labeled (this last comment applies to Figure 2).

4- In section 3.1, the authors showed the effects of cholesterol and its metabolite 27OHC in BC cell proliferation and invasion (Figure 1). In Section 3.2, the authors revealed that cholesterol effects on BC cell proliferation and migration are mediated by 27OHC since the effects were abolished with the enzyme CYP27A1 was silenced (Figure 2). Interestingly, in the discussion section, the authors indicated that both findings were previously reported (lines 330 to 337, refs. 10 and  14).  This referee recommends including this information in the results section and clearly detail to the reader where the novelty of the findings of the current manuscript lies. To the best of the understanding of this referee, the novelty lies in the fact that ER-beta would be the mediator of these effects (which is addressed in sections 3.3 and 3.4 corresponding to Figures 3 and 4). So, this referee encourages the authors to revise the results sections structure in order to clarify the novelties of their findings.

5- The authors use publicly available datasets to validate their observations. In the Methods section, the title indicated that they used the METABRIC dataset (line 142) but the description of the analysis revealed that they used the TCGA dataset (lines 144 and 158). In the results section, again they mentioned the METABRIC dataset when they described the results (lines 252, 281, and 306), but the figure legends indicated that they used the TCGA dataset (lines 263, 294, and 317). It is not clear to this reviewer which one of the two datasets was used in this study. Besides, it would be interesting to show that both datasets show similar results. Therefore, this reviewer encourages the authors to show the results from both datasets.

Minor concerns:

In the Methods section, please note that the description corresponding to “siRNA transfections” (Line 102) and “Trans-well cell migration/invasion assay” (Line 133), do not have their corresponding number ID.

Author Response

We thank the reviewer for their helpful comments.

Reviewer 1

1-In the Introduction the authors indicate (line 38): “Tumours expressing progesterone receptor (PR), estrogen receptor (ER) and/or the epidermal growth factor receptor 2 (HER2) are defined as hormone receptor-positive breast cancers, and triple-negative breast cancers (TNBC) lack PR, ER or HER2 expression [4]”. This referee believes that a more accurate definition of HR-positive BC should be included in the manuscript. Indeed, HR-positive breast cancers are the ones displaying ER and/or PR, independently of HER2 overexpression. There is a group of HR+ BCs that is HER-2 negative (HR+/HER2-: Luminal A- and Luminal B-like) and another group of HR+ BCs that are HER2-positive. On the other hand, HER2-positive BCs could be either HR-positive or HR-negative.

We have now clarified the subtypes in more detail (lines 38-43) as follows: ‘There are five main subtypes of breast cancer. Luminal A and B are both hormone receptor (estrogen-receptor (ER) and/or progesterone (PR)-receptor) positive, with luminal A being positive, and luminal B negative for HER2. Normal-like breast cancers are similar to luminal A, but with a slightly worse prognosis. Triple-negative/basal-like breast cancer is hormone receptor and HER2 negative, and HER-2 enriched tumours are HER2 positive and hormone receptor negative.

2- This reviewer found that the English language, grammar, and style of the Introduction and Discussion sections are adequate and engaging. However, a detailed revision of the English language and style is required in the Results section. Particularly, there are some words that are not usually used in scientific articles such as the word “negated”, whose meaning is not clear to this referee. Besides, this reviewer recommends the authors include a brief description of the aim and the BC models used in the first section of results (section 3.1).

The word ‘negated’ has been changed to ‘blocked’ as suggested (lines 223, 225, 242, 268, 294,363,371,399, 409). A short sentence indicating the aim and the cell lines used in section 3:1 has now been added (lines 185-187)- “We compared the effects of LDL (0-100μg/ml) on cell proliferation in an ERα-positive cell line, MCF-7 and in an ERα-negative breast cancer cell line, MDA-MB-231.” We have amended the results section titles to better reflect the findings (lines 183, 238, 264, 290, 325).

 3- This reviewer suggests that some modifications in the figure's style and figure legends would be friendlier to the readers. It would be interesting to include the cell line name in each panel.

The cell line name has now been added to the relevant panels in every figure.

Please include the WB images and their densitometric analysis in the same panel (Figure 1 panels I-L; Figure 5 panels C and D; Figure 6 panels A-D).

The format of the figures has been changed accordingly in figures no. 1, 5 and 6.

Regarding the images corresponding to invasion assays (Figure 1 panels E to H; Figure 2 panels B and E; Figure 3 panels C and D; Figure 4 panels A-C; Figure 5 panel B), please clearly indicate at which region from the image at 20X the inset corresponds.

Thank you for this comment. We have now overlaid the enlarged 20X inset onto the specific area of the original picture and have added a scale bar (100uM) for further clarity.

Please note that the statistic symbols are missing in: Figure 1 panels A, B, C, E, F, G, H, J, and L; Figure 2 all panels; Figure 3 panels A, C, D, and E; Figure 4 panels A-D; Figure 5 all panels.

Apologies for this oversight, we have now ensured that the statistical symbols are visible on all the graphs.

Regarding Figure Legends, please refer to the conventional style of description of the panels in which each panel ID is indicated before the description of the panel, and also, check that the panels are described consecutively as they are mentioned and labeled (this last comment applies to Figure 2).

The figure legends have been amended accordingly.

4- In section 3.1, the authors showed the effects of cholesterol and its metabolite 27OHC in BC cell proliferation and invasion (Figure 1). In Section 3.2, the authors revealed that the cholesterol effects on BC cell proliferation and migration are mediated by 27OHC since the effects were abolished with the enzyme CYP27A1 was silenced (Figure 2). Interestingly, in the discussion section, the authors indicated that both findings were previously reported (lines 330 to 337, refs. 10 and 14). This referee recommends including this information in the results section and clearly detail to the reader where the novelty of the findings of the current manuscript lies. To the best of the understanding of this referee, the novelty lies in the fact that ER-beta would be the mediator of these effects (which is addressed in sections 3.3 and 3.4 corresponding to Figures 3 and 4). So, this referee encourages the authors to revise the results section’s structure to clarify the novelties of their findings.

We thank the reviewer for this suggestion and have amended the text to highlight work that is confirmatory and data which are the key, novel findings gleaned from establishing these published models.

  • We have modified the end of the introduction to make it clearer what we have undertaken in the paper (Lines 95-99).

‘Herein, we established models in which we confirmed the effect of cholesterol and its metabolite, 27OHC on breast cancer cell proliferation, migration, invasion, and the abundance of epithelial-to-mesenchymal (EMT) markers. Using these models, we determined novel associations of molecules involved in cholesterol metabolism with the IGF/EGF axes and their dependency on ER-α and/or -β.

  • We modified the introductory sentences to sections 3.1 and 3.2.

3.1: We firstly confirmed the effects of LDL (0-100μg/ml) on cell proliferation in an ERα-positive cell line, MCF-7 and established that it also had similar effects on the proliferation of an ERα-negative breast cancer cell line, MDA-MB-231

3.2: We next confirmed that the effects of LDL on proliferation were mediated by CYP27A1 (enzyme responsible for the rate-limiting step in 27-hydroxycholesterol biosynthesis).

5- The authors use publicly available datasets to validate their observations. In the Methods section, the title indicated that they used the METABRIC dataset (line 142) but the description of the analysis revealed that they used the TCGA dataset (lines 144 and 158). In the results section, again they mentioned the METABRIC dataset when they described the results (lines 252, 281, and 306), but the figure legends indicated that they used the TCGA dataset (lines 263, 294, and 317). It is not clear to this reviewer which one of the two datasets was used in this study. Besides, it would be interesting to show that both datasets show similar results. Therefore, this reviewer encourages the authors to show the results from both datasets.

We included the METABRIC and TCGA datasets because both show similar results, which increases the validity of the observations as indicated by the reviewer. We have amended the text accordingly to make this clear (line 27, section 2:6, line 180, 271, 285, 305, 320, 343, 413, 432, 437).

Minor concerns: In the Methods section, please note that the description corresponding to “siRNA transfections” (Line 102) and “Trans-well cell migration/invasion assay” (Line 133), do not have their corresponding number ID.

The corresponding IDs have now been added on lines 114 and 116 respectively.

Reviewer 2 Report

The paper from Mashat et al. focuses on the effects of LDL and its metabolite, 27-OHC, on breast cancer progression and considers the possibility of the association with IGF/EGF axes and the involvement of ER-α and ER-ß in the induction of cell proliferation, migration, and invasion.

The aim of this manuscript is of interest; the introduction and the discussion are well organized, and the reading is fluent.

However, I have some concerns regarding the presentation of the results and the images.

-The histograms of figures 1, 2, 3, 4, 5 never show the statistical analysis. Please, uniform these figures with figure number 6, in which statistical analysis is correctly reported with the asterisks.

-In the results section, I recommend not to indicate, for each result, (p <0.0…) but to insert asterisks in the histograms when the results are significant.

-To make the figures much easier to read, show the cell type name used in each figure/panel.

-Could you please provide some theoretical explanation of the use of 24h incubation period for MCF7 and 6h for MDA-MB231 in trans-well cell migration/invasion assay? I can imagine, but it is necessary to report this concept in the manuscript.

Moreover, in the trans-well cell migration/invasion assay preliminarily fixing the cells on the filter, did you remove the cells that did not migrate or invade? The cells and gel in the upper compartment of the insert should be removed by gently wiping the upper side of the membrane with a cotton swab. Only the cells on the underside of the insert membrane are those that have migrated. This step is fundamental to avoid artifacts. Please explain, in the materials and methods section, how you proceeded.

-Page 3, line110: what does means 2Q?

-The citation Gao et al. 2013 reported on page 4, line 147 does not compare in the list of the references. Please, add it.

-Page 6, line 227: reference cited in the main text is incorrect, and I think you refer to Figure 3E and not to Figure 1E.

-page 14, line 410: the certified TNBC cell line used in this manuscript is negative for ERα, PR, and HER2, and not negative for “HER2 overexpression”, as you have indicated.

-Reference number 49 in the list at the end of the manuscript is not correctly indicated.

Author Response

Reviewer 2

We thank the reviewer for their helpful comments.

-The histograms of figures 1, 2, 3, 4, 5 never show the statistical analysis. Please, uniform these figures with figure number 6, in which statistical analysis is correctly reported with the.

Apologies for this oversight, we have now ensured that the statistical symbols are visible on all the graphs.

-In the results section, I recommend not to indicate, for each result, (p <0.0…) but to insert asterisks in the histograms when the results are significant.

Thank you for this recommendation. The histograms have been amended accordingly.

-To make the figures much easier to read, show the cell type name used in each figure/panel.

The cell type name has been added to appropriate panels of each figure.

-Could you please provide some theoretical explanation of the use of 24h incubation period for MCF7 and 6h for MDA-MB231 in trans-well cell migration/invasion assay? I can imagine, but it is necessary to report this concept in the manuscript.

Thank you for raising this point- we have included the rationale for this in the text as follows: The different incubation times used to assess cell migration and invasion reflected the differing invasive potential and aggressiveness of the two cell lines. (Line, 156-157).

Moreover, in the trans-well cell migration/invasion assay preliminarily fixing the cells on the filter, did you remove the cells that did not migrate or invade? The cells and gel in the upper compartment of the insert should be removed by gently wiping the upper side of the membrane with a cotton swab. Only the cells on the underside of the insert membrane are those that have migrated. This step is fundamental to avoid artifacts. Please explain, in the materials and methods section, how you proceeded.

The methodology for the trans-well cell migration/invasion assay has been extended for clarification as follows: ‘The cells in the filter were fixed using 4% PFA, and the cells in the upper compartment of the insert were removed by gently wiping the upper side of the membrane with a cotton swab, then cells on the underside were fixed, stained with 0.05% CV and permeabilised. (Line 158-160)

-Page 3, line110: what does means 2Q?

We apologise, this was a typographical error, that has now been deleted.

The citation Gao et al. 2013 reported on page 4, line 147 does not compare in the list of the references. Please, add it.

Thank you for noticing this omission, the reference has now been included in line 169 and in the reference list (ref.30).

-Page 6, line 227: reference cited in the main text is incorrect, and I think you refer to Figure 3E and not to Figure 1E.

 Thank you for spotting this error our reference to figure 1E has now been changed to figure 3E.

-page 14, line 410: the certified TNBC cell line used in this manuscript is negative for ERα, PR, and HER2, and not negative for “HER2 overexpression”, as you have indicated.

Thank you we have removed this incorrect terminology (Line, 424)

-Reference number 49 in the list at the end of the manuscript is not correctly indicated.

-Thank you-this reference Bin et al has now been referenced correctly in the list (now ref 53).

Reviewer 3 Report

The topic of the manuscript presented by Dr. Mashat and colleagues is interesting. However, it seems unfinished. The experiments are also well performed but preliminary. It does not add a good in-depth analysis.

They use two cellular lines and observe that proliferation and migration induced by 27OHC are triggered by ER alpha and ER beta respectively but in two different cell types. What about the migration of MCF-7 and proliferation of MDAMB231?

In addition, there is not an advancement in mechanistic insight. What are the pathways involved? These biological effects are driven by genomic or non-genomic-effects? 

My major concerns follow:

-Since the authors investigate the role of Er beta, they should add details about its variants (Endocrines 20212(3), 356-365; https://doi.org/10.3390/endocrines2030033) in the introduction or discussion section. 

-A paragraph related to siRNA approach and migration and invasion assays should be added separately.

-Antibodies codes should be added in Materials and Methods.

-It is not clear if in migration/invasion assays the authors add an inhibitor of proliferation o serum starvation conditions. 

-Figure 1: in addition to the proliferation assay reported the authors should add also another approach (FACS analysis, BrdU incorporation). In addition to the western blot presented, the authors should add also pictures (photos) in which they show EMT. Furthermore, the authors should specify if the cells that they are using are of epithelial or mesenchymal origin.

-Figure 2: The authors should use an internal GFP control, for verifying that the cells transfected (so silenced) are not migrating or proliferating. 

-Figure 3: ER-alpha has a role in the proliferation of MCF-7 cells. What is the pathway involved? In the legend, the authors write "27 hydroxycholesterol interacts with the ER-α ". How? Probably, "interact" is not the correct word.

-Figure 4: Also, in this case, ER-beta promotes migration and invasiveness of Triple-negative breast cancer, but how? Through what pathway?

Furthermore, the authors assess that ER-alpha is involved in cell proliferation and ER-beta in cell migration but in two different types of cells.

At this point what is the common thread that links 27OHC?

Have the authors analyzed the Androgen receptor that is involved in such effects, if is it linked to 27OHC? Or if there is a crosstalk between the steroid receptors? 

  • The authors write: "We found that the addition of LDL increased the proliferation and migration of MDA-MB-231 TNBC cells that was accompanied by an increase in the production of IGF-I and the abundance of the IGF-IR. These effects of LDL 
    were blocked in the presence of an IGF-1R tyrosine kinase inhibitor" Ok. At this point, the authors propose the idea of a feedback loop (not look, please correct). An autocrine loop is described also for other growth factors, such as NGF. The authors should discuss this autocrine loop (https://doi.org/10.3389/fcell.2021.676568). A spheroid model in which the authors cultivate their cells in 3D models for a long period (10 or 15 days) with the use of a neutralizing antibody could be appreciated.

Minor revision: Please, correct typos in the text

Author Response

Reviewer 3

They use two cellular lines and observe that proliferation and migration induced by 27OHC are triggered by ER alpha and ER beta respectively but in two different cell types. What about the migration of MCF-7 and proliferation of MDAMB231?

Thank you for the above comment, we also found that 27OHC increased the migration of MCF-7 cells and this was independent of ER- alpha (shown in figure 3C), and that 27OHC had no effect on the the growth of MDA-MB-231 (figure 1D).

In addition, there is not an advancement in mechanistic insight. What are the pathways involved? These biological effects are driven by genomic or non-genomic-effects? 

We thank the reviewer for this useful comment. We appreciate that we have not fully delineated the complete signalling pathway by which cholesterol and its metabolites are exerting their effects on proliferation, migration, and invasion, but we have identified a novel role of the ER-beta in their migratory effects and that this is associated with modifications in growth factor signalling pathways, suggesting a positive feedback loop. We have tried to highlight these points a little more in the text, in addition to trying to summarise our thoughts on mechanism in the summary as follows:

‘In summary, our data elaborates a mechanism to support the clinical studies suggesting the link between obesity and high cholesterol with an increased risk of breast cancer progression: cholesterol promotes cell proliferation, migration, and invasion and we show novel data to indicate that ERβ is central to the effects of LDL/27OHC on invasion and migration. We suggest that a positive feedback loop exists in response to cholesterol and its metabolites in breast cancer cells, that involves increased tyrosine kinase receptor phosphorylation and ER activation and subsequent nuclear translocation, where they upregulate target genes (including IGF-I, IGF-IR and EGFR). Delineating this suggested signalling pathway may identify novel opportunities for optimizing current breast cancer treatment regimens.

We also added some additional signalling data as a supplementary figure 1- to indicate that in MCF-7 cells, LDL increases levels and activation of the IGF-I1R that is blocked by an IGF-1R receptor tyrosine kinase inhibitor.

My major concerns follow:

- Since the authors investigate the role of Er beta, they should add details about its variants (Endocrines 2021, 2(3), 356-365; https://doi.org/10.3390/endocrines2030033) in the introduction or discussion section. 

Thank you for this suggestion, the following information has been added: Additionally, some subtypes of TNBC are responsive to estrogens [9]. ERβ has five different isoforms [7]. Different ER-β variants are expressed in TNBC; thus, ER-β1 variant works as a tumour suppressor. While ER-β2 and β5 seem to act as pro-oncogenes in TNBC [9]. Furthermore, ERβ1 is commonly expressed more than the other ER-β2 and 5 isoforms. The roles of the ER-β variants still remain unclear in TNBC. (line 51-56).

- A paragraph related to siRNA approach and migration and invasion assays should be added separately.

The approach to siRNA has been added as follows: ‘Lyophilised siRNA was resuspended with an appropriate amount of siRNA suspension buffer to make a 20μM stock solution, which was aliquoted and stored at -20°C. The concentration of the siRNA in the transfection master-mix depended on its required final concentration. The stock cells were trypsinised when 70-80% confluent, counted and then seeded into 6-well plates (0.1X106/cells in 800µl /well of GM). 200 µl of the transfection master-mix were added gently drop-wise. The cells were then incubated for 24 hours before changing the media to SFM for a further 24 hours at 37°C in 5% CO2. The transfection efficiency was assessed by western immunoblotting’ (lines 120-127).

The methodology for the trans-well cell migration/invasion assay has been extended for clarification as follows: ‘The cells in the filter were fixed using 4% PFA, and the cells in the upper compartment of the insert were removed by gently wiping the upper side of the membrane with a cotton swab, then cells on the underside were fixed, stained with 0.05% CV and permeabilised (Line 158-160)

- Antibodies codes should be added in Materials and Methods.

These details have been added to section 2:4.

- It is not clear if in migration/invasion assays the authors add an inhibitor of proliferation of serum starvation conditions. 

-We do not add an inhibitor of proliferation, because the duration time for the migration and invasion are less than 24hours for MCF-7 cells and 6 hours for the MDA-MB-231 cells; over these time frames no changes in cell number occur.

- Figure 1: in addition to the proliferation assay reported the authors should add also another approach (FACS analysis, BrdU incorporation). In addition to the western blot presented, the authors should add also pictures (photos) in which they show EMT. Furthermore, the authors should specify if the cells that they are using are of epithelial or mesenchymal origin.

We thank the reviewer for their suggestions. We did not perform an additional measure of cell proliferation as we predominantly used this method to just confirm previous work indicating the effects of LDL and 27OHC.

We have not included photographs of the cells undergoing EMT but have as requested indicated in the text the epithelial/mesenchymal origin of the cells.

The cell lines that were used have been specified as follows: ‘To confirm the effects of 27OHC and LDL on the abundance of EMT markers we used both MCF-7 (epithelial) and MDA-MB-231 (mesenchymal) breast epithelial cells. line 198. We also indicated this in the methods (line 108).

- Figure 2: The authors should use an internal GFP control, for verifying that the cells transfected (so silenced) are not migrating or proliferating. 

I thank the reviewer for their suggestion- we have used other suggested controls for siRNA such as, negative control siRNAs. We used RT-PCR and western blotting to check for the presence of mRNA and protein and ensure the efficiency of the silencing. We did not use GFP tagging as we were concerned that whilst feasible could interfere with RNA binding.

- Figure 3: ER-alpha has a role in the proliferation of MCF-7 cells. What is the pathway involved?

We had added a sentence to the discussion highlighting some of our previous findings regarding the mechanism-our previous paper showed that in MCF-7 cells ER alpha is phosphorylated, enters the nucleus where it can upregulate target gene expression.

‘We previously showed in MCF-7 cells that activation of the ER alpha resulted in its phosphorylation and translocation to the nucleus where it upregulated expression of target genes [39]’. (line 373)

In the legend, the authors write "27 hydroxycholesterol interacts with the ER-α ". How? Probably, "interact" is not the correct word.

  • We agree that ‘interact’ is not the correct description- it has been rephrased to; ‘Effects of 27- hydroxycholesterol on cell growth, migration/invasion in the presence or absence of the ERα (line 254).

- Figure 4: Also, in this case, ER-beta promotes migration and invasiveness of Triple-negative breast cancer, but how? Through what pathway? Furthermore, the authors assess that ER-alpha is involved in cell proliferation and ER-beta in cell migration but in two different types of cells.

We thank the reviewer for this comment. We did use two different cell lines. MCF-7 cells, that have both ERs and the MDA-MB-231 cells, that only express ER-beta. The literature seems to suggest that the relative levels of these ERs in the cells dictates their response. We found that ER-alpha was not involved in MCF-7 cell migration and invasion, but that ER-beta could be a candidate. We, therefore, switched to MDA-MB-231 cells, that only express ER-beta, to try to prove a role for ER-beta in these effects. We have tried to assimilate our data better in our discussion summary and suggest a potential mechanism (as outlined above), but that also acknowledges there is still work to be done to completely define this signalling pathway, in cells that express both or only one of the receptors.

Have the authors analyzed the Androgen receptor that is involved in such effects, if is it linked to 27OHC? Or if there is a crosstalk between the steroid receptors? 

The reviewer raises an interesting point, that was beyond the scope of this paper. However, we have added a paragraph about AR and its links to 27OHC in the discussion as follows: ‘A recent study suggested an association between ER-β and the androgen receptor (AR) receptor, where ER-β and AR act together in TNBC to modulate cell function [9]. Further work would be required to confirm the relative contribution of AR to the actions of 27OHC on breast cancer cells (line 390-393). Also, we have added a paragraph in the introduction as follows:’ Recently, the role of ER-β in relation to the androgen receptor (AR)and its ability to mediate the receptor-mediated effects was investigated and it was suggested that the AR could be a marker in TNBC [9] (line 56-60).

The authors write: "We found that the addition of LDL increased the proliferation and migration of MDA-MB-231 TNBC cells that was accompanied by an increase in the production of IGF-I and the abundance of the IGF-IR. These effects of LDL were blocked in the presence of an IGF-1R tyrosine kinase inhibitor" Ok. At this point, the authors propose the idea of a feedback loop (not look, please correct).

Thank you for this suggestion where ‘look’ has been changed to ‘loop’ (line, 299).

 An autocrine loop is described also for other growth factors, such as NGF. The authors should discuss this autocrine loop (https://doi.org/10.3389/fcell.2021.676568).

Thank you for this suggestion which has been changed (line, 400-402) as follows: ’An autocrine loop has also been described for other growth factors, such as nerve growth factor (NGF) which modulates an autocrine loop through the activation of tyrosine kinases, that may contribute to the sustained growth and aggressiveness of TNBC

 A spheroid model in which the authors cultivate their cells in 3D models for a long period (10 or 15 days) with the use of a neutralizing antibody could be appreciated.

Thank you for this suggestion which has been added as an excellent suggestion for future work as follows:’. It would be interesting in future to examine these effects in more physiologically relevant models, including colony formation and growing cells in 3D (line, 402)

Minor revision: Please, correct typos in the text

Minor revisions have been made to address any typos.

Round 2

Reviewer 1 Report

Comments to the authors

This reviewer really appreciates the authors’ efforts to answer this reviewer’s comments. The revised version of the manuscript improves the clarity and quality of the article.

However, this reviewer still has some concerns.

In the response to this reviewer comment #1, the authors included a more detailed definition of BC subtypes. This reviewer again encourages the authors to revise their description because there are molecular subtypes defined by genome-wide techniques and clinical subtypes that are the surrogate subtypes determined by immunohistochemistry that are used in the clinic to guide therapeutic techniques. Although there is a high correlation between molecular and clinical subtypes, they do not always match. Luminal A and B are molecular subtypes. In the clinic, luminal A and B molecular subtypes mainly belong to the HR-positive and HER2-negative clinical subtypes (AnnOncol 2017;28:1700-12). I would not mention that luminal A are HER2+ and luminal B are HER2-, because indeed it is generally the other way around (Lum B can be HER2+, J Clin Oncol 2012; 31:203-209). From a clinical perspective, HR-positive breast cancers are the ones displaying ER and/or PR, independently of HER2 overexpression. There is a group of HR+ BCs that is HER-2 negative (HR+/HER2-: Luminal A- and Luminal B-like) and another group of HR+ BCs that are HER2-positive. On the other hand, HER2-positive BCs could be either HR-positive or HR-negative.

-   In the response to this reviewer comment #5, the authors included both METABRIC and TCGA datasets results in the manuscript, further validating their observations.

In Figure 4, the authors include panel E (ERbeta mRNA from TCGA), panel F (ERbeta mRNA from METABRIC), and panel G indicates CYP27A1 but it is not indicated from which cohort since Figure Legend 4 does not include panel G. In Figure Legend 6 the authors indicate that panel D contains TCGA data for IGF-IR, ki67, and EGFR mRNA, but the figure only includes 2 graphs (IGF-IR and ki67) under panel D and the EGFR graph is at the bottom of panel C. Besides, in Figure Legend 6E, the authors indicate that they show EGFR, CYP27A, and IGF-1 mRNA data, but the image only shows EGFR and IGF-1R. Please revise the above mention panels and legends to make them clear for the readers.

Author Response

Reviewer Rebuttal

We thank the reviewer for carefully checking our revisions and appreciate the time they have invested in our paper. We apologise for the oversights in the figure legends.

In the response to this reviewer comment #1, the authors included a more detailed definition of BC subtypes. This reviewer again encourages the authors to revise their description because there are molecular subtypes defined by genome-wide techniques and clinical subtypes that are the surrogate subtypes determined by immunohistochemistry that are used in the clinic to guide therapeutic techniques. Although there is a high correlation between molecular and clinical subtypes, they do not always match. Luminal A and B are molecular subtypes. In the clinic, luminal A and B molecular subtypes mainly belong to the HR-positive and HER2-negative clinical subtypes (AnnOncol 2017;28:1700-12). I would not mention that luminal A are HER2+ and luminal B are HER2-, because indeed it is generally the other way around (Lum B can be HER2+, J Clin Oncol 2012; 31:203-209). From a clinical perspective, HR-positive breast cancers are the ones displaying ER and/or PR, independently of HER2 overexpression. There is a group of HR+ BCs that is HER-2 negative (HR+/HER2-: Luminal A- and Luminal B-like) and another group of HR+ BCs that are HER2-positive. On the other hand, HER2-positive BCs could be either HR-positive or HR-negative.

We thank the reviewer for this explanation and have inserted a revised, simpler but more accurate description of the breast cancer subtypes. Lines 38-42

-   In the response to this reviewer comment #5, the authors included both METABRIC and TCGA datasets results in the manuscript, further validating their observations.In Figure 4, the authors include panel E (ERbeta mRNA from TCGA), panel F (ERbeta mRNA from METABRIC), and panel G indicates CYP27A1 but it is not indicated from which cohort since Figure Legend 4 does not include panel G. In Figure Legend 6 the authors indicate that panel D contains TCGA data for IGF-IR, ki67, and EGFR mRNA, but the figure only includes 2 graphs (IGF-IR and ki67) under panel D and the EGFR graph is at the bottom of panel C. Besides, in Figure Legend 6E, the authors indicate that they show EGFR, CYP27A, and IGF-1 mRNA data, but the image only shows EGFR and IGF-1R. Please revise the above mention panels and legends to make them clear for the readers.

We apologise for these errors and have amended accordingly.

Figure 4 legend now includes a description of panel G.

Figure 6: The panels representing the TGCA and METABRIC data have been labelled correctly (D, E & F) and described accurately in the legend.

Reviewer 2 Report

The authors responded consistently to my requests. I have no additional comments.

Author Response

We thank the reviewer for their positive response.